# Infrared Image Caption Based on Object-Oriented Attention

**DOI:** 10.3390/e25050826

**Published:** 2023-05-22

**Authors:** Junfeng Lv, Tian Hui, Yongfeng Zhi, Yuelei Xu

**Affiliations:** Institute of Unmanned System Research, Northwestern Polytechnical University, Xi’an 710072, China

**Keywords:** infrared image caption, domain transfer object detection, adaptive weighting module, object oriented attention

## Abstract

With the ongoing development of image technology, the deployment of various intelligent applications on embedded devices has attracted increased attention in the industry. One such application is automatic image captioning for infrared images, which involves converting images into text. This practical task is widely used in night security, as well as for understanding night scenes and other scenarios. However, due to the differences in image features and the complexity of semantic information, generating captions for infrared images remains a challenging task. From the perspective of deployment and application, to improve the correlation between descriptions and objects, we introduced the YOLOv6 and LSTM as encoder-decoder structure and proposed infrared image caption based on object-oriented attention. Firstly, to improve the domain adaptability of the detector, we optimized the pseudo-label learning process. Secondly, we proposed the object-oriented attention method to address the alignment problem between complex semantic information and embedded words. This method helps select the most crucial features of the object region and guides the caption model in generating words that are more relevant to the object. Our methods have shown good performance on the infrared image and can produce words explicitly associated with the object regions located by the detector. The robustness and effectiveness of the proposed methods were demonstrated through evaluation on various datasets, along with other state-of-the-art methods. Our approach achieved BLUE-4 scores of 31.6 and 41.2 on KAIST and Infrared City and Town datasets, respectively. Our approach provides a feasible solution for the deployment of embedded devices in industrial applications.

## 1. Introduction

Image information entropy can be applied to the field of image processing. An image can be considered a two-dimensional array of pixels in image processing. Entropy can measure the complexity or the amount of information in an image, i.e., how much information is present in the image. For example, in a black-and-white image, if each pixel has only two possible values (black or white), the entropy value will be meager because the information in the image is elementary.

In recent years, with the development of computer technology, various intelligent applications are no longer limited to server-side deployment. The demand for deploying functions such as autonomous object detection, target tracking and positioning, and image captioning on embedded devices is increasing. However, embedded devices often need to consider power supply and consumption issues, making it difficult to directly deploy many large models. Therefore, it is necessary to find a method that is easy to deploy, requires less computing resources, and can maintain good performance.

Image captioning involves describing the contents of an image in sentences, bridging the gap between image processing and natural language processing. It is a critical component of intelligent applications. The encoder–decoder structure holds a dominant position in the field [1,2]. These methods consist of a cascade of convolutional layers that form an image encoder. The image is then passed through a pre-trained convolutional neural network to carry out the coding process, with the one-dimensional feature vector extracted at the network’s fully connected layer. A recurrent neural network then forms a decoder to fit the one-dimensional feature vector to the embedded annotated sentences.

In industrial applications, infrared sensors are preferred over visible light sensors, as they can capture images in all weather conditions and are resistant to meteorological interference. Infrared sensors measure physical characteristics and can capture images from any object emitting infrared light, provided it has a temperature higher than absolute zero. Unlike visible light sensors, they can be used in low-light conditions and at night, making them particularly useful in extreme weather conditions such as fog, rain, and snow [3]. They have broad application prospects for night tasks. Therefore, studying image captioning based on infrared images is of significant importance, and has broad practical application value for the deployment of embedded devices. However, there are still many challenges to be addressed, which will be discussed in detail below.

Lacking precision relevant to the established object. In our preliminary experiments, we found that pre-trained models tend to describe the different content in an image, including the relationships and states between different objects. While the results generated in this way are diverse, it is difficult to obtain descriptive results that are highly relevant to the established object. For example, as can be seen from Figure 1, when multiple objects are present, the annotation often describe the image from multiple perspectives, where the results of (3) are typically considered more complete and accurate for describing the behavior of the main object in the image. However, in actual tasks, there are always some objects that we focus on and some that can be ignored. Thus, different methods tend to generate varying results due to the uncertainty of global information. To achieve this effect, some researchers have utilized the semantic attention mechanism [4] to align the feature map regions and embedded words. The attention modules tend to learn each word and corresponding image region; the low-level semantic features help the model to select the most essential and relevant image regions when generating a caption. However, for nouns, there are specific image areas that match them, but are inaccurate when directly corresponding verbs, prepositions, and adjectives to image areas. This may result in the generation of redundant information during feedback processing. Moreover, without the high-level information guidance, the model may prioritize visually appealing image regions that are semantically unrelated to the main subject of the image. Therefore, when considering the requirements of practical tasks, we usually want the model to focus on certain predefined objects. It is crucial to incorporate high-level information to guide the model in generating descriptions relevant to the predefined objects.

Lacking domain adaptiveness. The performance of a model depends heavily on the support of datasets. Using pre-trained models and fine-tuning their parameters through transfer learning is a widely adopted technique. However, this approach often encounters overfitting caused by sample imbalancing. For instance, in object detection tasks, we frequently use certain object types as targets for transfer learning. The transfer results on the dataset usually exhibit good accuracy. However, during testing, we often observe high false positive rates, i.e., similar features are incorrectly identified as the designated targets. Image captioning faces similar problems. When the images and annotations in the dataset lack diversity, the training tends to favor specific high-frequency vocabulary, resulting in very similar descriptive results and overfitting issues. Introducing more diverse and comprehensive datasets can alleviate this problem. However, most open-source datasets are focused on visible light images. The same object has different features in visible light images and infrared images. In Figure 2, some visible light and infrared images of the same scene and targets are presented, including daytime and nighttime. From the figure, it can be seen that the objects in the visible light images have relatively different textures and colors, while the infrared images contain the contour features of the objects. The different feature distributions of visible light and infrared lead to domain gaps, which seriously affect their performance. On the other hand, most pre-trained models are trained on massive visible light datasets with many instance-level annotations [5]. When implementing these pre-trained models, it is important to follow a potential rule that ensures the testing and training images are distributed in the same feature space. For instance, if we set the infrared images as the target domain and visible-light images as the source domain, pre-trained models cannot be implemented directly. Certainly, we can also increase the amount of data in the target domain by re-labeling, but this is a very time-consuming and labor-intensive task. Therefore, in order to avoid overfitting caused by sample imbalancing in practical applications, existing open-source datasets should be used to expand the feature space in the target domain and achieve the knowledge transfer. It is essential to enhance the diversity and richness of the target domain data as much as possible.

Based on the above discussion, we followed the encoder–decoder structure and proposed an infrared image caption method. We extracted image features and high-level semantic information based on YOLOv6 [6], and treated them as the encoder. Then, we used LSTM to implement decoding. The primary contributions of this paper are as follows:

(1) Firstly, we enhanced the knowledge transfer process by leveraging the approach outlined in reference [7]. We refined the selection process for pseudo-labels, which has resulted in a more robust detector capable of effectively adapting to changes in images.

(2) Secondly, we introduced the object-oriented attention module, which combines high-level semantic information and image features to weigh the proportion of each component in the multi-level features during word generation. This approach guides the model to generate descriptions that are specifically related to the predefined object.

(3) We provided an infrared dataset of streets in both towns and cities and conducted extensive experiments across multiple datasets to validate the effectiveness of our proposed method. The results from our simulations demonstrate the efficacy of our approach.

## 2. Related Work

Image captioning refers to the task of generating natural language descriptions for images, and it has attracted significant attention in recent years. In this section, we will discuss several commonly used methods, which can be classified into two categories: Neural Network-Based and Transformer-Based image captioning.

### 2.1. Neural Network Based

Encoder–decoder models based on convolutional neural networks and recurrent neural networks are the most widely used structures in image captioning. Much of the research in this field has focused on improving the quality of generated text [8], enhancing model robustness, and addressing challenges such as multilingualism and multimodality. To overcome these issues, researchers have proposed numerous innovative methods, such as utilizing attention mechanisms [9,10] to focus on different regions of an image. There are two main types of attention mechanisms used in image captioning: soft attention and hard attention. Soft attention assigns weights to all image regions at each decoding step, with weight values ranging between 0 and 1. On the other hand, hard attention attends to a specific region of the image at each decoding step, using one-hot encoding. Hard attention is computationally less expensive and typically employs Monte Carlo sampling to estimate gradients and perform backpropagation during training. The attention mechanism transforms the global image features extracted by CNNs into regional features. Before generating each word of the caption, the attention mechanism computes the relevance of each image region to the word to be generated. Then, the features of the highly relevant regions are selected and passed to the decoder to guide the generation of the next word.

The introduction of the attention mechanism enables image captioning models to focus on specific image regions. However, mechanically assigning each vocabulary word to a corresponding region in the image may not be the most effective approach for guiding sentence generation. This is because some words, such as “of” and “the”, do not necessarily correspond to specific regions in the image, and forcing them to do so can lead to unnecessary computational waste. Therefore, some researchers have studied the topic-guided method. Yu et al. [11] proposed an image captioning framework based on the topic-oriented method, which uses a multi-label classifier to select the main topics and establish a hierarchy in the generated sentences. Similarly, Zhu et al. [12] proposed the topic-guided attention method, in which the most frequent topic words are extracted from candidate sentences and used as the basis for sentence generation. Chen et al. [13] proposed the topic words embedding method to represent the high-level semantic information. Image captioning methods based on the neural network have achieved good results. Researchers can further improve the model performance by using attention mechanisms or topic-oriented mechanisms. However, in practical applications, there is a need for further research investigating how to effectively guide the model to generate accurate descriptions of given objects.

### 2.2. Transformer Based

The Transformer [14] was originally applied to natural language processing and has been widely used in the field of computer vision in recent years, achieving good results, such as Transformer-based object detection [15] and classification [16]. Based on the encoder–decoder structure, Wang et al. [17] introduced the CapFormer, a method used to model historical words in image captioning tasks. CapFormer uses cross-focus layers to interact with image features and improve the accuracy of the generated captions. Liu et al. [18] proposed the RS image captioning, which leverages the transformer architecture to enhance image features using the bitemporal feature differences from two temporal images. As for the decoder, the multilayer aggregated transformer [19] has been proposed to utilize the extracted feature. Cornia et al. [20] proposed an image captioning model based on a Meshed-Memory Transformer. Both the encoder and decoder of the model use transformer. The encoder encodes image regions and their relationships, while the decoder utilizes low-level and high-level visual relationships to generate the output. Zhou et al. [21] proposed the Unified VLP based on the shared multi-layer transformer for both encoding and decoding.

Transformer-based models have demonstrated increasingly powerful performance. From neural networks to Transformers, researchers have been working hard to find better ways to improve the correlation between image and text features. Recently, some researchers have also been studying a unified framework for cross-modal learning, integrating tasks such as image captioning, image classification, and language modeling, such as mPLUG [22] and OFA [23]. To address the problem of Transformers requiring the flattening of grid features before being fed into the Encoder, which destroys the relative position information in the 2D grid, and the issue of some words (such as “with”) having little relevance to the image, RSTNet [24] proposed the Grid-Augmented module and Adaptive Attention module to improve the visual representation of relative position information and measure the contribution of visual information to generating captions. Currently, the methods for improving model structure are becoming increasingly mature. However, due to the large amount of self-attention, its computational resource requirements are higher than those of convolutional neural networks and recurrent neural networks. Additionally, the Transformer requires a large amount of training data to achieve certain results, and is highly dependent on training data [25]. Furthermore, as the Transformer does not have the hidden state of recurrent neural networks, it is very sensitive to the positional information of the input sequence.

The above content highlights that despite significant progress in research, there are still limitations and challenges in practical deployment.

One such challenge is the semantic diversity in images and description statements. Images are rich in semantic information, including various objects, scenes, and actions. It is challenging to convey all of this information in a single description result, and different descriptions of an image from different perspectives can lead to varying results. While most research focuses on guiding models to adapt to complex sentence content in Ground Truth, without clear object guidance, models often combine unrelated objects or content, leading to imprecise results. Therefore, there is a need for further research to improve the model’s ability to produce highly relevant results to the object.

Another significant challenge is how to enhance the adaptive domain of the model. While a good performance assumes that training and testing datasets share the same origin, real-world applications often have different data sources, especially in scenarios in which unmanned devices use dual-light devices for all-weather detection. Thus, further research should explore knowledge transfer methods that utilize existing resources to improve the model’s performance.

## 3. Methods

In this section, we will provide a detailed description of our approach. Firstly, we introduce the domain transfer method that we used to perform domain adaptation for high-level information extraction. Secondly, we introduce the object-oriented attention module, which enables the generation of words that are explicitly linked to objects detected in an image. By combining the global information provided by the image feature with local information obtained from image regions and object classes, we guide the model through the word generation process. An overview of our method can be found in Figure 3. Firstly, we use domain transfer method to transfer the detector, obtaining the images of the approximate domain through the generative adversarial network, and fine-tune the detector based on this. Then, we use the fine-tuned detector to obtain pseudo-labels in the target domain and finally fine-tune the detector based on the pseudo-labels in the target domain and the labels in the approximate domain, which makes the detector more adaptive to infrared images. Given an infrared image, we can obtain the image features, object features, and corresponding categories based on the detector. These features will be combined with word features to form the training data, and then the model will be trained using the object-oriented attention method with adaptive weighting module, making the model tend to generate statements related to objects.

### 3.1. Domain Transfer Method

The main differences between visible light images and infrared images lie in their low-level features, such as edges, textures, and colors. Visible light images typically exhibit clear, well-defined objects, while infrared images tend to have lower contrast and only show contour features. In light of these differences, it is important to reduce the distance between the distributions of these two domains. Our approach is based on the semi-supervised learning method, which is illustrated in Figure 4, we improved the pseudo-label learning process based on the domain similarity loss, it helps select the optimal result and is regarded as a pseudo-label. The visible light images can be considered the source domain $D_{S}$ and the infrared images can be considered the target domain $D_{T}$. We also implemented the CycleGAN [26] to alleviate the domain gap.

The adversarial neural network processing involved transferring the two different domain images I and J:(1)G:I→J
(2)F:J→I
where G and F are convolutional neural networks. An unsupervised adversarial network was created by combining the two mapping relationships, and trained until it reached equilibrium. The domain transfer between the source domain $D_{S}$ and target domain $D_{T}$ was achieved using this theoretical method. The main process is as shown below:(3)$$i\in R^{H_{1}\times W_{1}\times 3}$$
(4)$$D_{S}=\left(i,b,c\right)$$
(5)$$D_{T}^{\prime }=\left(i^{\prime },b,c\right)$$
where i represents the visible light image and $i^{\prime }$ represents the visible light image transferred by the adversarial network. $H_{1}$ and $W_{1}$ represent the height and width of visible light images, respectively. c∈C refers to the categories and $b\in R^{4}$ represents the bounding boxes of the source domain $D_{S}$.

After the aforementioned operations, each transferred visible light image $i^{\prime }$ in the target domain $D_{T}$’ was distributed as an instance-level annotation. The detector was fine-tuned on $D_{T}^{\prime }=\left(i^{\prime },b,c\right)$ by means of transfer learning. Then, the target domain images in $D_{T}$ were input into the fine-tuned detector, producing coarse outputs with a list of confidence scores for each category. We selected the top 5 results with the highest confidence scores as candidate annotations. To determine the best candidate, we cropped the top 5 results based on their respective coordinates and utilized the detector’s backbone to reduce the domain gap and identify the most suitable result.
(6)$${D_{T}}_{top5}=\left({conf}_{top5},b,c\right)$$

We utilized the backbone and added the fully connected layer to infer the transferred image patch $f_{s}$, extracting the corresponding vector $v_{s}$. We also inferred the candidate patches of the target domain $f_{t_{m}},m=1,2,3,4,5$, and extracted the corresponding vectors $v_{t_{m}},m=1,2,3,4,5$. The vectors $v_{s}$ and $v_{t_{m}}$ were then utilized to calculate the domain similarity loss ${loss}_{ds}$ with the formal expression as follows.
(7)$$v_{s}=\Phi \left(f_{s}\right)$$
(8)$$v_{t_{m}}=\Phi \left(f_{t_{m}}\right)$$
(9)$$conf\left(v_{s},v_{t_{m}}\right)=\frac{v_{s}\cdot v_{t_{m}}}{{\left\|v_{s}\right\|}_{2}{\left\|v_{t_{m}}\right\|}_{2}}$$
(10)$${loss}_{ds}=\frac{1}{m}\sum_{i=1}^{m}{\left(p_{i}-{conf}_{i}\right)}^{2}$$
where Φ is the classifier, $conf\left(v_{s},v_{t_{m}}\right)$ represents the cosine distance of $v_{s}$ and $v_{t_{m}}$. $conf=\left[{conf}_{1},{conf}_{2},\ldots ,{conf}_{5}\right]$ represents the domain similarity and $p=\left[p_{1},p_{2},\ldots ,p_{m}\right]$ is the similarity label; in our experiment, we set this to 1 and 0. Finally, the highest confidence result was selected and considered as a pseudo-label, which can be formulated as:(11)$$j\in K^{H_{2}\times W_{2}\times 3}$$
(12)$$D_{T}=\left(j,c\right)$$
(13)$${D_{T}}_{best}=\left(p_{best},j,b,c\right)$$
where j represents the infrared image. $H_{2}$ and $W_{2}$ represents the height and width, respectively. $p_{best}$ denotes the result with the highest domain similarity and ${D_{T}}_{best}$ refers to the pseudo-labels of the target domain. In the final step, the detector was fine-tuned on U, which consists of $D_{T}^{\prime }$ and ${D_{T}}_{best}$.

### 3.2. Object-Oriented Attention Module

Following domain transfer, the detector could be employed for infrared images. This section will outline the process of generating image captions from object information and features. Object classes serve as low-level intuitive features, whereas visual features represent high-level semantic features of a deep model. To fuse these low-level and high-level features, a multi-stage feature fusion module has been proposed. The key function of this module is to determine the weight proportion of each part of the multi-level features for word generation.

Given the infrared image V, let $r_{j}=\left\{r_{1},r_{2},\ldots ,r_{n}\right\}$ represent the object regions of input image V and $c_{j}=\left\{c_{1},c_{2},\ldots ,c_{n}\right\}$ represent the classes of corresponding object regions, we first reshaped the feature map by flattening its width and height. This can be formulated as:(14)$$v=f\left(V\right)$$
(15)$$v^{\prime }=Flatten\left(v\right)$$
(16)$$r_{j}^{\prime }=Flatten\left(r_{j}\right)$$
where $r_{j}^{\prime }$ represent the object-oriented semantic features, f demotes the backbone, and v is the encoded image feature extracted by the backbone.

Then, we concatenated the image feature $v^{\prime }$ and the $h_{t-1}^{c}$ of LSTM to form the input of attention-LSTM and update the state of the hidden layer.
(17)$$x_{t}^{a}=concat\left(h_{t-1}^{c},v\right)$$
(18)$$h_{t}^{a}=att_{LSTM\left(x_{t}^{a},h_{t-1}^{a}\right)}$$
where $x_{t}^{a}$ is the concatenated feature, $att_{LSTM}$ represents the attention-LSTM. The $h_{t}^{a}$ represents the updated hidden layer state.

The main function of the adaptive weighting module is to determine the object region features and corresponding classes, which is the crucial part of object-oriented attention, as shown in Figure 5.

In the first stage, we concatenated the $r_{j}=\left\{r_{1},r_{2},\ldots ,r_{n}\right\}$ and $c_{j}=\left\{c_{1},c_{2},\ldots ,c_{n}\right\}$; the activated feature vector $\alpha_{1}$ was obtained by the full connection layer and hyperbolic tangent function layer. We computed the normalized weight $\alpha_{1}^{\prime }$ using *SoftMax*. The weighted feature $r_{j}^{1}$ and $c_{j}^{1}$ was obtained via multiplication with normalized weight $\alpha_{1}^{\prime }$. The processing can be formulated as follows:(19)$$v_{1}=concat\left(r_{j},c_{j}\right)$$
(20)$$\alpha_{1}=\tanh\left(W_{v}^{1}v_{1},W_{h}^{1}h_{t}^{a}\right)$$
(21)$$\alpha_{1}^{\prime }=softmax\left(\alpha_{1}\right)$$
(22)$$\left\{\begin{array}{c} r_{j}^{1}=\sum_{j=1}^{n}\alpha_{1}^{\prime }r_{j}\\ c_{j}^{1}=\sum_{j=1}^{n}\alpha_{1}^{\prime }c_{j} \end{array}\right.$$

In the second stage, the weighted feature $r_{j}^{1}$ and $c_{j}^{1}$ obtained in the first stage were concatenated. The structure of the full connection layer, hyperbolic tangent function layer and *SoftMax* was also included. The whole process was the same as shown in the first stage, and can be formulated as follows:(23)$$v_{2}=concat\left(r_{j}^{1},c_{j}^{1}\right)$$
(24)$$\alpha_{2}=\tanh\left(W_{v}^{2}v_{2},W_{h}^{2}h_{t}^{a}\right)$$
(25)$$\alpha_{2}^{\prime }=softmax\left(\alpha_{2}\right)$$
(26)$$\left\{\begin{array}{c} r_{j}^{2}=\sum_{j=1}^{n}\alpha_{2}^{\prime }r_{j}^{1}\\ c_{j}^{2}=\sum_{j=1}^{n}\alpha_{2}^{\prime }c_{j}^{1} \end{array}\right.$$

Lastly, we concatenated the refined features of object regions $r_{j}^{2}$ and classes $c_{j}^{2}$, and input them into LSTM to align with embedded words, thus achieving decoding. The processing is shown in Figure 6, and the calculation formula is as follows:(27)$$x_{t}^{c}=concat\left(h_{t}^{a},r_{j}^{2},c_{j}^{2}\right)$$
(28)$$h_{t}^{c}=LSTM(h_{t-1}^{a},x_{t}^{c})$$
(29)$$p\left(y_{t}\left|y_{t-1}\right.\right)=softmax\left(W_{p}h_{t}^{c}+b_{p}\right)$$
where $p\left(y_{t}\left|y_{t-1}\right.\right)$ represents the conditional probability of each word being generated at t.

## 4. Experiment

In this section, we provide a detailed description of the experimental methodology used in our study. We evaluated our proposed method on three different datasets: Pascal VOC2012 [5], KAIST [27], and Infrared City and Town. Following the conventional image caption annotation method, we annotated the KAIST [27] and Infrared City and Town. The datasets are described below:

Pascal VOC2012 [5]: The dataset consists of 20 object categories, and the training and validation sets contain a total of 11,530 images with 27,450 ROI-annotated objects. This dataset has been widely used for object recognition and detection tasks.

KAIST [27]: The dataset is a multispectral pedestrian dataset that contains visible-light and infrared images. The dataset has a total of 103,128 dense annotations and 1182 distinct objects. We annotated the infrared images in this dataset, and for each image, we provided five sentences of manual description.

Infrared City and Town: This dataset was built by us, and contains three main object categories: airplane, car, and pedestrian. We captured the images using infrared equipment in the streets of both cities and towns, including various lighting and weather conditions such as sunny, cloudy, and rainy. We also annotated this dataset using five sentences per image.

Our experiments were divided into two parts:

(1) We validated the effectiveness of domain transfer method on the basis of two sets of experiments: Pascal VOC2012 [5] to KAIST [27] and Pascal VOC2012 [5] to Infrared City and Town.

(2) Through combination with the detector, we validated the effectiveness of the infrared image caption method on KAIST [27] and Infrared City and Town, respectively.

### 4.1. Experimental Details and Metrics

All models were implemented using Python 3.6 and PyTorch 1.9 and trained using an NVIDIA 2080Ti GPU.

For domain transfer evaluation, we used a pre-trained detector. The detector was optimized with SGD and included L2 regularization. The learning rate was set to 0.001. In addition, data augmentation techniques such as random crop, random flip, and random brightness adjustments were used. We evaluated the detector using Precision (*Pr*), Recall (*Re*), F1 score (*F*1), and mean Average Precision (mAP), which are widely used in detection tasks. They can be calculated as follows:(30)$$Pr=\frac{TP}{TP+FP}$$
(31)$$Re=\frac{TP}{TP+FN}$$
(32)$$F1=\frac{2\times Pr\times Re}{Pr+Re}$$
where *TP*, *FP* and *FN* represent the true positive, false positive and false negative, respectively.

For the infrared image caption experiments, we converted all annotations to lower case and removed function words, non-numeric, and non-alphabetic characters that did not provide information in the descriptions. We then counted the occurrences of the remaining words and used words that appeared more than three times to build a dictionary, which was converted to a one-hot vector. A total of 8532 words were used to train the image caption model. The learning rate was set to 0.001 and the batch size was set to 8. The input layer had a dimension of 2048 and the output layer had a dimension of 1024. The class features, region features, and word embedding vectors were set to 512 dimensions. The maximum length of a generated sentence was set to 15. We used publicly available metrics, including BLEU-1, BLEU-2, BLEU-3, BLEU-4, METEOR, and CIDEr. The BLEU-n score is widely used for machine translation, and can be calculated as follows:(33)$$\mathrm{BP}=\left\{\begin{array}{cc} 1,\; & c>r\\ e^{1-\frac{r}{c}},\; & c\leq r \end{array}\right.$$
(34)$$BLEU=BP\times e^{\left(\sum_{n=1}^{N}\omega_{n}\log p_{n}\right)}$$
where r refers to the reference annotation, and c refers to the candidate sentence. $\omega_{n}$ and $P_{n}^{B}$ are weights and precision of n-grams. N=1,2,3,4, $\omega_{n}=\frac{1}{N}$.

METEOR is based on word-to-word matching scores, and the calculation is as follows:(35)$$P=\frac{m}{c}$$
(36)$$R=\frac{m}{r}$$
(37)$$F_{mean}=\frac{P\times R\times 10}{9\times P+R}$$
(38)$$METEOR=F_{mean}\times \left(1-\frac{0.5\times ch}{m}\right)$$
where P and R are caption precision and recall, respectively. m represents the number of matched words. *ch* refers to chunk, a series of contiguous and identically ordered matches between the generated captions and the annotation.

CIDEr evaluates the consensus between a candidate sentence and annotation and calculates the frequency of n-grams in a candidate sentence based on term frequency–inverse document frequency (TF-IDF). The weighting processing can be formulated as follows:(39)$${CIDEr}_{n}\left(c_{i},r_{i}\right)=\frac{1}{m}\sum_{j}\frac{g^{n}\left(c_{i}\right)g^{n}\left(r_{ij}\right)}{\left\|g^{n}\left(c_{i}\right)\right\|\left\|g^{n}\left(r_{ij}\right)\right\|}$$
(40)$$CIDEr\left(c_{i},r_{i}\right)=\sum_{n=1}^{N}\omega_{n}{CIDEr}_{n}\left(c_{i},r_{i}\right)$$
where $r_{i}$ refers to the reference annotation and $c_{i}$ refers to the candidate’s sentence. $g^{n}$ is the vector consisting of all n-grams of length *n*.

### 4.2. Quantitative Results of the Domain Transfer Method

In our approach, we use detectors to extract image features and high-level semantic information. Therefore, in this section of the experiment, we began by validating the effectiveness of the domain transfer method. We selected widely used models such as FasterRCNN [28], SSD [29], YOLOv4 [30], and YOLOv6 [6] as comparison methods and chose the common categories “Car and pedestrian” in datasets as the transfer target. The backbone for FasterRCNN [28] and SSD [29] is Resnet50, and YOLOv4 [30] and YOLOv6 [6] both use the m-size.

First, we conducted the experiment of Pascal VOC2012 [5] to KAIST [27], where we used Pascal VOC2012 [5] as the source domain and KAIST [27] as the target domain to observe the performance changes. Table 1 shows that YOLOv6m+DT (domain transfer) achieved precision, recall, F1, and mAP of 58.47%, 60.33%, 59.39%, and 62.47%, respectively. Compared to the method without domain transfer, precision, recall, F1, and mAP increased by 35.33%, 37.76%, 36.54%, and 38.90%, respectively. Similar improvements were observed in other detectors, such as YOLOv4m+DT and YOLOv4m, which increased mAP by 35.47%. The two-step detector also showed improvement, with FasterRCNN_Resnet50+DT achieving an mAP about 32.85% higher than FasterRCNN_Resnet50.

In another set of experiments, we employed Pascal VOC2012 [5] on the Infrared City and Town dataset. We used Pascal VOC2012 [5] as the source domain and Infrared City and Town as the target domain, and we observed similar patterns. As shown in Table 2, we obtained precision, recall, F1, and mAP of 81.64%, 80.93%, 81.28%, and 83.16%, respectively. Compared to the method without domain transfer, precision, recall, F1, and mAP increased by 50.82%, 49.76%, 50.29%, and 48.19%, respectively. The SSD_Resnet50 had poorer performance, but also achieved an improvement of 47.11% of mAP. The detectors performed better in Infrared City and Town than in KAIST [27]. After analyzing the results, we found that many objects in Infrared City and Town have a relatively transparent background compared to KAIST [27]. Therefore, it is easier to distinguish the foreground from the background.

In both experiments, both single-step and two-step detectors improved when combined with the domain transfer method. The feature distribution of the detectors switched from visible light to infrared after domain transfer using transfer learning, making the detectors more adaptable to infrared object features. Based on the above discussion, the experimental results demonstrate the effectiveness of our domain transfer method. We achieved domain adaptation through the adversarial model and fine-tuned the detectors through pseudo-learning. Ultimately, the detector trained on visible-light datasets demonstrated good domain adaptability on infrared datasets.

### 4.3. Quantitative Evaluation Results for Infrared Image Caption on KAIST

In this section of the experiment, we compare our method with several existing image caption models. We divided them into four types: (1) Neural network-based methods, such as Vgg16+RNN, Vgg16+LSTM, Neural Baby Talk [2], Google NIC [1], and Noc [8]; (2) Transformer-based methods, such as M2 Transformer [20], Unified VLP [21] and RSTNet [24]; (3) Attention-based methods, such as soft attention [9], semantic attention [10], Yu et al. [11], OGA [12], C-LSTM [24], and our proposed method; and (4) Multimodal-based methods, such as mPLUG [22] and OFA [23]. The structural composition of the model in each of the methods described above is presented in Table 3. All methods were set up with consistent basic settings. All experiments were conducted offline using two NVIDIA 2080ti GPUs. The learning rate for the encoder was set to 0.001, and the learning rate for the decoder was set to 0.004. Both the encoder and decoder were optimized using the Adam method, and training was conducted for 150 epochs. The same training set was used for all experiments, and no data augmentation was performed in this part. The maximum length for generating words was set to 15.

Table 4 shows the corresponding infrared image captioning performances on KAIST [27]. Our proposed method obtained BLUE-4, METEOR and CIDEr scores of 32.6%, 26.8%, 111.2%, respectively. These were the highest scores on both datasets for all metrics. Our method outperformed neural network-based methods such as Google NIC [1] and Noc [8] by 6.5% and 4.7% in BLUE-4. Moreover, compared to other attention-based methods such as soft attention [9], semantic attention [10], Yu et al. [11], and OGA [12], our object-oriented attention method performed better, utilizing local object regions and high-level information fully, and demonstrating more than 3.0%, 2.1%, 6.1%, and 6.7% improvements in METEOR, respectively. While our method achieved significant advantages compared to the Neural network- and Attention-based methods, it achieved relatively similar results to those achieved by the Transformer-based methods and the Multimodal-based methods. For instance, compared to RSTNet [24] (ResNext152), our method achieved an improvement of 3.7% and 2.9% for BLEU-4 and METEOR, respectively. In addition, our method outperformed M2 Transformer [20] and Unified VLP [21] by 4.0% and 3.4% in BLUE-4, respectively. Additionally, compared to mPLUG [22] and OFA [23], our method scored 1.2% and 2.1% higher in METEOR. Although the performance of the transformer-based methods and the multimodal-based methods was very close to ours, our method has a simpler structure, requires less computational resources, has faster inference speed, and is more compatible with deployment on embedded platforms. Our proposed method also achieved a higher score than the others in terms of CIDEr, illustrating the similarity of the sentence to the ground truth. This suggests that our method can clearly state the objects in an image. These observations indicate that image captioning based on object regions’ semantics improves model performance significantly compared to describing global semantics. Our proposed method eliminates redundant information from other image areas and focuses more on the object.

### 4.4. Quantitative Evaluation Results for Infrared Image Caption on Infrared City and Town

Similar to the previous section, we continued to use the methods mentioned earlier for testing on the self-built dataset. Table 5 shows the comparison experiment results on Infrared City and Town. The performance of each evaluation metric improved. Our proposed method obtained BLUE-4, METEOR and CIDEr scores of 42.2%, 36.9%, 127.3%, respectively. Compared to C-LSTM [24], BLUE-1, BLEU-2, BLEU-3 and BLEU-4 increased by 2.4%, 3.6%, 2.8% and 4.6%, respectively. Additionally, compared to Google NIC [1], our method outperformed it by 7.4%, 9.8%, 6.4% and 7.0% (BLUE-1, BLEU-2, BLEU-3, and BLEU-4, respectively). For the classical Neural Talk method [2], our method outperformed the other models by almost 11.3% and 16.0% in terms of METEOR and CIDEr, respectively. The proposed object-oriented attention method also obtained a better performance than other attention methods, such as soft attention [9] and semantic attention [10]: by more than 7.3% and 5.1% for BLUE-4 and more than 7.8% and 6.2 for METEOR. As in the previous section, the results achieved by the methods based on the transformer and the multimodal were very close to those achieved using our method. For example, our method outperformed mPLUG [22] and Unified VLP [21] by 1.6% and 2.6% for BLUE-4, and 1.3% and 2.2% for METEOR, respectively. In addition, compared to RSTNet [24] (ResNext101) and RSTNet [24] (ResNext152), METEOR improved by 3.0% and 2.3%, and BLEU-4 improved by 3.6% and 2.8%, respectively. However, our method still maintained a good performance in terms of CIDEr. The adaptive weighting module can fully exploit the potential relevance of the class feature, region feature, and embedded word vector, and can improve the performance of image captioning by enabling interaction between each component of the visual region feature. The introduction of this module breaks down the isolation of object regions and high-level information in the image, revealing the semantic relevance of each image region by comprehensively considering the location and content relevance. Regions with high semantic relevance can assist each other in generating words, which effectively improves the model’s ability to understand the image content. Based on the above analysis, the object-oriented attention method can guide the allocation of weights and provide more useful information for the text generation model. It is easy to conclude that the image captioning method based on object-oriented attention can enhance the model’s ability to understand regional relationships and improve the description generation performance.

### 4.5. Quantitative Evaluation Results and Embedded Platform Porting

Figure 7 displays examples of image caption results with their corresponding annotations. We selected methods from each of the four types as the object of comparative simulation testing, and these are: Google NIC [1], Semantic attention [10], M2 Transformer [20] and mPLUG [22]. Our method generates descriptions that are relevant to the objects in the images. The annotations attempt to describe the image’s complex semantic content. For instance, in the second infrared image from left to right, the annotations include “pedestrians”, “cars”, “trees”, and “buildings”. The sentences describe these objects separately, but the crucial aspects of the foreground are the “pedestrians” and “cars”. The other elements can be considered as background. In our opinion, the description of an image’s content should prioritize the foreground. Our method can describe both objects based on the domain-transferred detector and object-oriented attention.

Moreover, for the last image from left to right, the annotations contain different topic sentences, such as “mountain”, “tunnel”, and “airplane”. Our method focuses on the object itself, accurately generating the most relevant description of the “airplane”. These examples demonstrate that our method can accurately describe the complex semantic information of an infrared image, and we have achieved similar performances on both examination datasets. Based on the above discussion, the qualitative results further support the effectiveness of the proposed method.

## 5. Conclusions

In this paper, a method is proposed for generating captions for infrared images based on object-oriented attention. Our approach involves two models: a detector and an LSTM. We first fine-tune the detector on visible-light images that have undergone style transfer. Then, we utilize the fine-tuned detector to acquire pseudo-labels on the target domain with image-level annotation. Lastly, we fine-tune the detector based on the pseudo-labels and visible-light images that have undergone style transfer to obtain the final detector. Notably, to address the feature differences between visible light and infrared images, we propose the domain similarity loss, which optimizes the selection process of the pseudo-label, expands the range of the target domain distribution, and improves the adaptability of the detector. The transferred detector enables the LSTM to select the most relevant regions in the foreground and eliminate redundant semantics, resulting in more accurate and robust descriptions. We also introduce an object-oriented attention module for the LSTM that uses object classes and regions as guiding information to align corresponding embedded words. The resulting descriptions are more accurate and robust due to the high-level information guidance. We conduct comprehensive experiments on two infrared datasets, and the results demonstrate the effectiveness of our approach. Furthermore, our approach is suitable for implementation on embedded devices, as it requires fewer resources and is convenient to deploy.

## Figures and Tables

**Figure 1 entropy-25-00826-f001:**
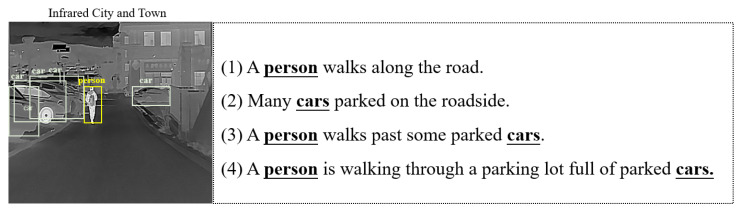
Different caption results are generated from different methods. (1) Generated from the baseline model [1]. (2) Generated from the baseline model [2]. (3) Represents our proposed method. (4) The Ground Truth.

**Figure 2 entropy-25-00826-f002:**
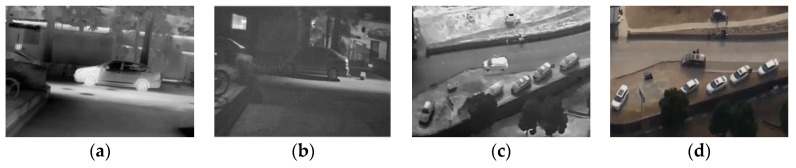
Infrared images and visible light images. (**a**) Infrared image; (**b**) visible light image (night); (**c**) infrared image; (**d**) visible light image (daytime).

**Figure 3 entropy-25-00826-f003:**
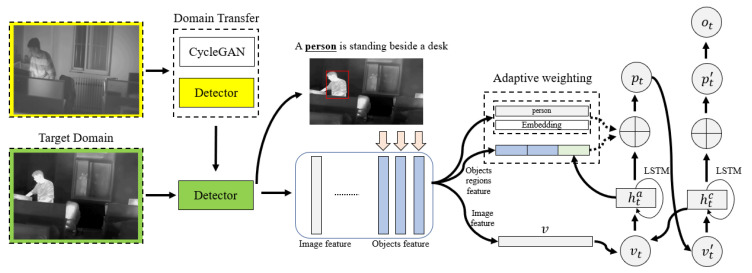
The overview of our proposed methods.

**Figure 4 entropy-25-00826-f004:**
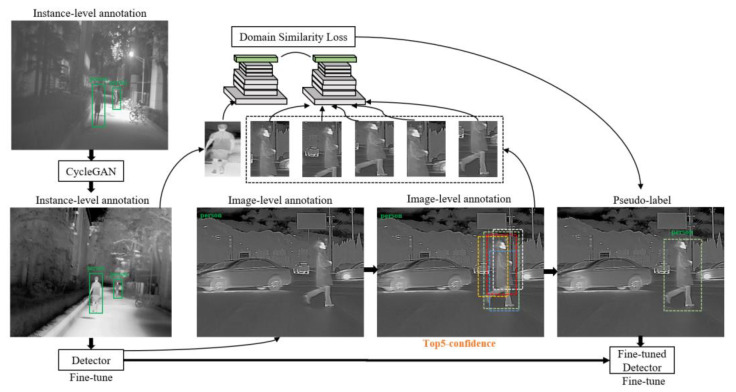
Domain transfer.

**Figure 5 entropy-25-00826-f005:**
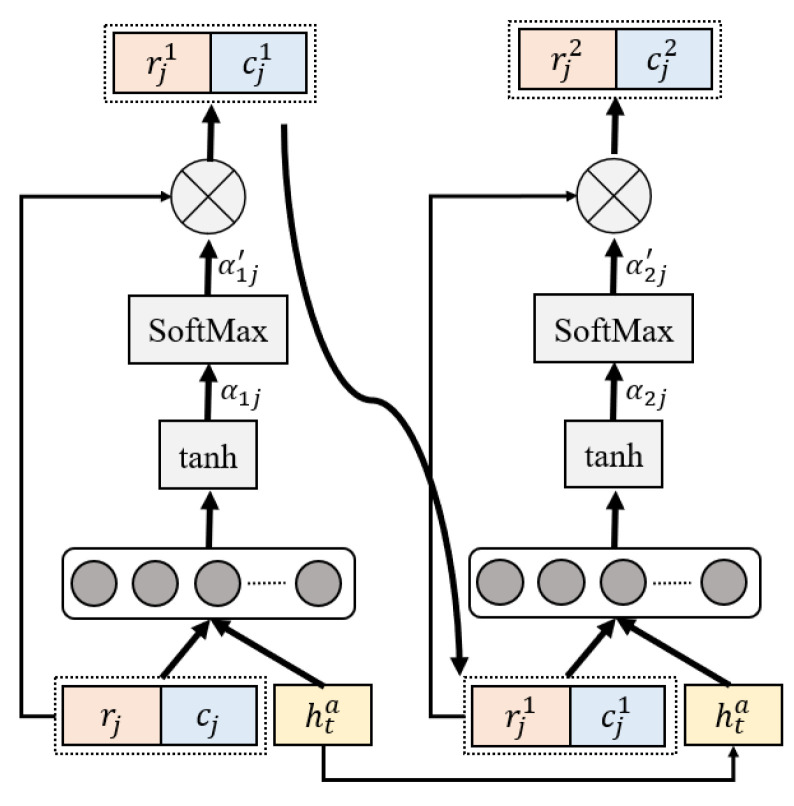
The adaptive weighting module.

**Figure 6 entropy-25-00826-f006:**
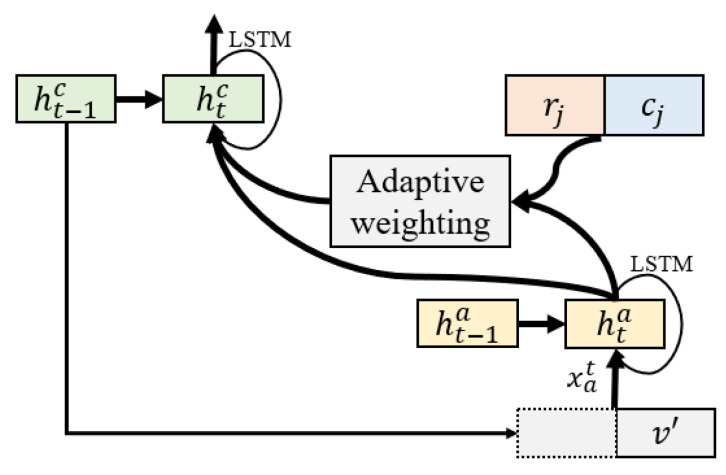
Overview of object-oriented attention.

**Figure 7 entropy-25-00826-f007:**
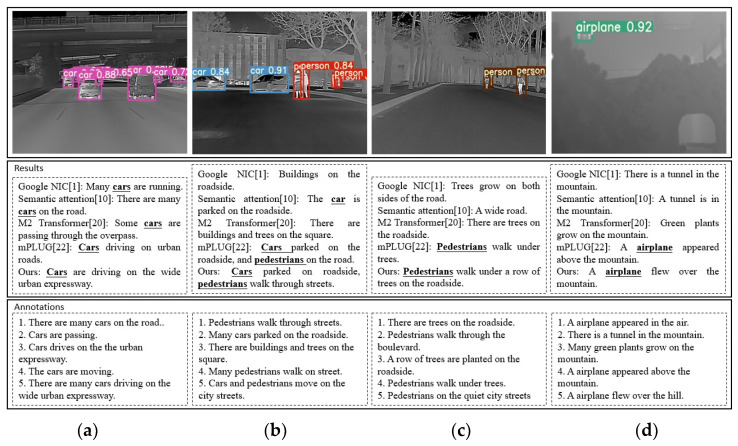
Infrared image captions results. (**a**) Urban scenery; (**b**) urban scenery; (**c**) rural scenery; (**d**) mountainous scenery.

**Table 1 entropy-25-00826-t001:** Detection results of Pascal VOC2012 to KAIST/%.

Method	Pr	Re	F1	mAP
FasterRCNN_Resnet50	18.72	20.67	19.65	21.82
FasterRCNN_Resnet50+DT	52.69	53.53	53.11	54.67
SSD_Resnet50	15.41	16.82	16.08	17.14
SSD_Resnet50+DT	49.32	50.81	50.05	51.36
YOLOv4m	20.21	21.76	20.96	22.17
YOLOv4m+DT	55.18	56.63	55.89	57.64
YOLOv6m	23.14	22.57	22.85	23.57
YOLOv6m+DT	58.47	60.33	59.39	62.47

**Table 2 entropy-25-00826-t002:** Detection results of Pascal VOC2012 to Infrared City and Town/%.

Method	Pr	Re	F1	mAP
FasterRCNN_resnet50	25.34	26.93	26.11	27.48
FasterRCNN_resnet50+DT	72.48	73.56	73.02	74.18
SSD_resnet50	21.47	21.36	21.41	22.02
SSD_resnet50+DT	66.21	64.52	65.35	69.13
YOLOv4m	28.46	30.27	29.34	33.28
YOLOv4m+DT	75.94	77.16	76.55	79.31
YOLOv6m	30.82	31.17	30.99	34.97
YOLOv6m+DT	81.64	80.93	81.28	83.16

**Table 3 entropy-25-00826-t003:** Structural composition of the models.

Method	Visual Feature Extraction	Caption Generation
Vgg16+RNN	Vgg16	RNN
Vgg16+LSTM	Vgg16	LSTM
Neural Baby Talk [2]	Faster RCNN_Resnet101	LSTM
Google NIC [1]	Inceptionv3	LSTM
Soft attention [9]	Vgg16	LSTM
Semantic attention [10]	Vgg19	LSTM
Noc [8]	Vgg16	LSTM
C-LSTM [24]	Vgg16	LSTM
M2 Transformer [20]	Faster RCNN_Resnet101	Transformer
Unified VLP [21]	Transformer	Transformer
RSTNet [24]	ResNext101	Transformer
RSTNet [24]	ResNext152	Transformer
Yu et al. [11]	Vgg19	LSTM
OGA [12]	Vgg16	LSTM
mPLUG [22]	Visual-Transformer	Transformer
OFA [23]	Transformer	Transformer
Ours	EfficientRep	LSTM

**Table 4 entropy-25-00826-t004:** Performance of the proposed model on the KAIST compared with other models/%.

Method	BLEU-1	BLEU-2	BLEU-3	BLEU-4	METEOR	CIDEr
Vgg16+RNN	55.7	46.3	37.3	22.7	18.4	94.1
Vgg16+LSTM	59.4	49.3	40.2	25.6	20.3	102.8
Neural Baby Talk [2]	58.6	48.2	39.6	23.1	19.6	98.7
Google NIC [1]	62.7	50.1	41.4	26.1	21.4	101.4
Soft attention [9]	65.9	53.6	43.2	28.7	23.8	104.9
Semantic attention [10]	66.3	53.7	43.8	29.1	24.7	108.1
Noc [8]	64.8	52.4	42.1	27.9	22.6	102.7
C-LSTM [24]	64.1	51.7	42.6	28.3	23.0	103.4
M2 Transformer [20]	65.8	52.1	41.9	28.6	23.7	104.7
Unified VLP [21]	66.1	52.8	42.7	29.2	24.3	105.2
RSTNet [24](ResNext101)	65.3	51.6	41.7	28.1	23.2	103.9
RSTNet [24](ResNext152)	66.0	52.4	42.3	28.9	23.9	104.9
Yu et al. [11]	61.2	50.3	40.6	25.7	20.7	100.3
OGA [12]	60.1	49.7	38.2	23.6	20.1	99.4
mPLUG [22]	67.1	54.5	43.8	31.2	25.6	108.6
OFA [23]	66.7	53.4	43.4	30.3	24.7	107.7
Ours	68.3	55.1	44.8	32.6	26.8	111.2

**Table 5 entropy-25-00826-t005:** Performance of the proposed model on the Infrared City and Town, compared with other models/%.

Method	BLEU-1	BLEU-2	BLEU-3	BLEU-4	METEOR	CIDEr
Vgg16+RNN	66.5	51.4	44.1	28.7	23.1	105.9
Vgg16+LSTM	68.3	53.1	46.7	30.2	24.3	107.7
Neural Talk [2]	67.2	52.8	46.1	30.8	25.6	111.3
Google NIC [1]	71.4	57.6	50.7	35.2	29.7	121.3
Soft attention [9]	72.1	58.4	50.2	34.9	29.1	119.4
Semantic attention [10]	75.9	62.7	53.8	37.1	30.7	119.8
Noc [8]	75.8	61.2	52.7	36.8	30.3	121.4
C-LSTM [24]	76.4	63.8	54.3	37.6	32.9	124.6
M2 Transformer [20]	76.8	64.1	54.7	39.2	34.3	124.7
Unified VLP [21]	77.1	64.5	55.1	39.6	34.7	125.1
RSTNet [24](ResNext101)	76.4	63.7	54.3	38.6	33.9	124.3
RSTNet [24](ResNext152)	77.2	64.4	54.9	39.4	34.6	124.9
Yu et al. [11]	71.3	57.6	50.5	35.3	29.6	119.3
OGA [12]	70.8	56.1	48.9	33.8	28.4	116.2
mPLUG [22]	77.8	66.7	55.7	40.6	35.6	125.8
OFA [23]	77.4	66.2	55.2	40.2	35.1	125.4
Ours	78.8	67.4	57.1	42.2	36.9	127.3

## Data Availability

The Pascal VOC2012 and KAIST datasets are openly available in a public repository. They can be downloaded at http://host.robots.ox.ac.uk/pascal/VOC/voc2012/, accessed on 10 May 2022. and https://github.com/SoonminHwang/rgbt-ped-detection/blob/master/data/README.md, accessed on 1 March 2022. The Infrared City and Town dataset is available on request from the authors. The data that support the findings of this study are available from the corresponding author upon reasonable request.

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
