# Peer review of "Infrared Image Caption Based on Object-Oriented Attention"

_entropy, 2023, doi:10.3390/e25050826_

Round 1

Reviewer 1 Report

The paper studies the well-known image captioning field. I believe that it is difficult to fully understand the novelty and contribution of the paper. These are the reasons:

1. The Related Studies section is weak. It does not describe what is the advantage of the proposed method compared with existing SOTA methods. What is interesting, the authors do not describe in the Related Studies the methods that were used in the experiments [35, 36, 37, 38].

2. Some of the representative papers in the field of image captioning are completely missing. I expect the authors to compare (both theoretically and experimentally) the proposed method with some of the SOTA methods found here:

https://paperswithcode.com/task/image-captioning

3. I have serious concerns regarding the proposed method's accuracy as well. Table 1 shows that the accuracy of the proposed method can achieve as high as 39% on the KAIST dataset even though it improves over the basic YOLO models.

4. Lastly, the paper contains so many grammatical mistakes, awkward and unfinished sentences. 

Author Response

We are very grateful for the suggestions made by the reviewers, which were very helpful for the manuscript. We have made a lot of revisions and improvements to the manuscript. Please see the attachment and review it again.

Reviewer 2 Report

Revise the article title to make it more concise.

The last part of the abstract should be rephrased to highlight the answer to your objectives

include the cross-domain in the keywords section

why used image from other reference , because the authors provided a reference number in thee caption of figure 1.

caption in figure 3 should be reduced, the explanation should be provided in the main context.

the caption of figure 5 should be together with their mage/figure 

the content of table 4 should be on the same page

section title (conclusion) should be on page 14

Provide a simulation/testing to have a comparative analysis on how it will highlight your concept against other existing concepts

Author Response

(The authors gave the same response as above.)

Round 2

Reviewer 1 Report

I appreciate the authors' point-to-point response. However, I still believe that there are several issues in the paper:

1. The authors included several papers in the experiments (though, I don't think these are SOTA). For a fair comparison, the implementation details of these papers are entirely missing. What backbone architecture did you use to implement them? Do they have a similar configuration as your method? Etc.

2. Regarding the accuracy of the proposed method. In the revised paper, the authors miraculously improved the accuracy (from 39% to 59% in the F1 score). However, neither the revised paper nor the authors' response file contains the reason how and why it suddenly gained improved accuracy. 

Author Response

We would like to express our gratitude once again to the reviewers for their valuable comments. Thank you for your review and dedication, which have been of great help in revising the paper. The revised content can be found in the attached file.

Reviewer 2 Report

1. Provide additional segment or insert another portion of image in figure 1 locating or highlighting the objects defined

2. On Figure 2, please insert letters (e.g., a, b, c, and d) to identify each image and provide a caption describing the type of image presented.

3. in the overview of the proposed method on figure 3, what will be the final terminal of the arrow-bar?

4. same goes in figure 7, provide letters to segment or individual identification of each image in this figure.

5. The concluding section should be expanded to provide a more comprehensive discussion, as the current focus is solely on the LSTM module

Author Response

(The authors gave the same response as above.)

Round 3

Reviewer 2 Report

It seem some of the images in the figures needs some enhancement

some portions of the contexts are not clearly describe the corresponding images/figures

provide additional keywords that are significantly describe your study

Author Response

We would like to express our gratitude once again to the reviewer for valuable comments. Thank you for your review and dedication, which have been of great help in revising the paper. The revised content can be found in the attached file.
